# Automotive Steel with a High Product of Strength and Elongation used for Cold and Hot Forming Simultaneously

**DOI:** 10.3390/ma14051121

**Published:** 2021-02-27

**Authors:** Fei Huang, Qiwei Chen, Hanlin Ding, Yongqiang Wang, Xiuting Mou, Jian Chen

**Affiliations:** 1School of Materials Science and Engineering, Anhui University of Technology, Maanshan 243002, China; maszhjs@126.com (F.H.); agdcqw@163.com (Q.C.); agdmxt@163.com (X.M.); 17301526993@163.com (J.C.); 2School of Iron and Steel, Soochow University, Suzhou 215006, China; dinghanlin@suda.edu.cn

**Keywords:** C–Mn–Cr automotive steel, cold forming, hot forming, high product of strength and elongation

## Abstract

A low-cost and easy-to-produce C–Mn–Cr automotive steel for both cold and hot forming is presented in this paper. The alloying element Cr was used to replace Mn in medium-Mn steel and instead of B in hot-formed steel, in order to achieve microstructure control and hardenability improvement, replacing the residual austenite-enhanced plasticization with multidimensional enhanced plasticization through multiphase microstructure design, grain refinement, and dispersion enhancement of second-phase particles. The products of strength and elongation for the cold-formed and hot-formed steel were 20 GPa·% and 18 GPa·%, respectively, while the tensile strengths were more than 1000 MPa and 1500 MPa, respectively. This new automotive steel was also characterized by good oxidation resistance. The mechanisms of strength and plasticization of the experimental automotive steel were analyzed.

## 1. Introduction

With energy consumption and environmental problems, automotive light-weighting has become an important part of the development of the automotive industry [1,2]. The most important part of light-weighting is the development of new-generation material for automobiles [3,4,5]. Despite the emergence of high-strength plastics and light metal materials for automobiles [6], research into and development of lightweight steel materials for automobiles has been of great importance to researchers for decades [7,8]. Research and development of new automotive steels is ongoing [9,10]. To date, steel has remained the most commonly used material in automotive structures due to its cost and full life-cycle environmental assessment, such that it is still considered irreplaceable [11,12].

The product of strength and elongation for automotive steel is the characteristic of the energy absorbed by an automobile in a collision event. Therefore, it provides a characterization of safety. The research and development of steel for automobiles requires not only high strength but also good plasticity for automotive components; thus, the development trend of the new-generation of automotive steels is to obtain high plasticity under ultra-high strength conditions [13]. In 2007, the United States put forward the concept of third-generation automotive steel, which requires that, under the tensile strength of 1000 MPa, the elongation reaches 30% for the key structural parts of the car; this remains the goal in the research and development of high-performance automotive steels.

The current research on high-performance steel for automobiles is mainly divided into two directions: steel for automobiles and for automotive components. The research and development of automotive steels is focused on quenching and partition (QP) treatment and medium manganese alloying [14,15], the basic principle of which is the use of residual austenite phase transformation-induced plasticity (TRIP), twin-induced phase transition (TWIP), or both to improve the plasticity of the steel [16,17]. QP980, produced by Baosteel (Shanghai, China), has a strength of over 980 MPa, an elongation of 20%, and a product of strength and elongation of nearly 20 GPa·%; however, the production process is complex to control [18]. The strength of medium manganese steels can range from 800 to 1200 MPa, with a strength and elongation product of 25 to 45 GPa·% [19]; however, automobile steel must be molded into automobile parts to realize its application. The hot forming process requires austenitization and phase transformation of the austenite, which inevitably leads to the destruction of the original residual austenitic microstructure. Therefore, automotive steels based on residual austenitic reinforcement and plasticization are currently only suitable for the cold forming of automotive parts. For ultra-high-strength steel plates, a series of problems relating to cracking, rapid mold wear, and elastic spring back often appears in the cold forming process. Therefore, the molding of 1000 MPa or greater automotive steels still has technical difficulties [20,21,22]. In order to resolve these technical problems, hot stamping technology, represented by ArcelorMittal (Luxembourg), has been vigorously developed. Parts formed through hot stamping are also the mainstream products of European and American automobile systems. However, although the commonly used hot-formed automotive components can achieve high strengths, their elongation rate is generally very low. For example, the typical hot-formed part of 22MnB5 has a strength of 1500 MPa but an elongation of only 6%. The product of strength and elongation is obviously low, the cold bending resistance is poor, and the anti-collision ability of the automobile body cannot meet the requirements of high-performance automobile parts [23,24,25].

In view of the above problems existing in the research and development of high -performance automotive components and automotive steel, a new composition design or automotive steel is proposed in this paper, utilizing multidimensional microstructure control [3,19]. The developed automotive steel has an excellent high product of strength and elongation and can be used with both cold forming and hot forming methods to produce high-performance automotive components.

## 2. Materials and Methods

For the experiment, C–Mn–Cr low-alloy steel was used; its main chemical composition is shown in Table 1 [26]. The steel was smelted in a 130kg vacuum induction furnace, cast into ingots of 150 mm × 150 mm × 700 mm, forged into 250 mm × 150 mm × 40 mm billets at temperatures between 1200 and 950 °C, and then hot rolled into 4.4 mm-thick steel plates by a two-high rolling mill with a 450 mm diameter of roll through 11 passes. The initial and final hot rolling temperatures were 1150 and 900 °C, respectively. After the hot rolling mill, the steel plates were kept at 650 °C for 1 h and cooled to room temperature. Finally, the hot-rolled plates were pickled and cold rolled for ten passes to 2 mm thickness by using a four-high rolling mill with a 400 mm diameter of the back-up roll and a 200 mm diameter of the working roll.

Subsequently, (1) the steel sheets were annealed (continuous annealing process to simulate actual production; the process route is shown in Figure 1) and then a tensile test was performed at room temperature; (2) the cold-rolled steel sheets were pressed into U-shapes (see Figure 2) using hot stamping presses, then a tensile test was performed at room temperature and the microstructure was observed. The hot-stamping process parameters were as follows: the heating temperature was 890 °C, maintained for 3 min; the die rate was 60 mm/s; the quenching time was 20 s; the holding pressure was 200 kN; and the cooling water pressure was 0.6 MPa.

The microstructure was observed with a ZEISS Axio Observer A1m optical metallurgical microscope (ZESSInc, Oberkochen, Germany) and a JSM-6610 scanning electron microscope (JEOL, Tokyo, Japan). Energy-dispersive X-ray spectroscopy (EDS) was performed using an Apollox energy spectrometer (EDAXInc, Mahwah, NJ, USA) at 40 kV. Electron backscattered diffraction (EBSD)was collected at a step size of 0.3 μm by an OXFORD HKL EBSD detector (OXFORD instrumentsInc, London, UK) at 20 kV, andthe corresponding data were processed withthe channel5 software. The piece samples, with sizes of 10 mm × 10 mm × 2 mm cut from hot-stamped steel, were abraded using silicon carbide abrasive paper to 2000# first, and then polished with diamond paste with a size of 1.5 μm, and after that polished by electrolyte. The solution for electrolytic polishing was 5% perchloric acid ethanol (volume fraction), the electropolishing voltage was 30 V, and the polishing time was 15 s. The tensile tests were carried out on a Z050 tensile testing machine (Zwick RoellInc, Ulm, Germany). X-ray diffraction (XRD) measurements were performed on the normal surface of rolled sheets using a Bruker D8 ADVANCE X diffractometer (BrukerInc, Karlsruhe, Germany) with Co-Kα radiation. A voltage of 35 kV and current of 40 mA were utilized. The diffraction angles were scanned from 45 to 115° in 2θ.

## 3. Results and Discussion

### 3.1. Microstructure and Properties of Cold-Rolled Sheets after Continuous Annealing

The tensile properties of cold-rolled sheets after continuous annealing (simulating actual production) are shown in Figure 3 and Table 2. It can be seen that the tensile strength, elongation, and the product of strength and elongation for the cold-rolled plate were 1013 MPa, 20.0%, and 20.3 GPa·%, respectively. These properties are related to the composition and process design of the experimental steel. These results are similar to those of commercial QP980 [18], but the experimental steel was produced through a conventional cold rolling and continuous annealing process without requiring any special facilities.

The microstructure of automotive steel after continuous annealing (simulating actual production) is shown in Figure 4. It can be seen that the fine microstructure consisted of multiple phases of bainite(B), residual austenite (RA), and second-phase particles (carbides). This multiphase, multidimensional microstructure provides excellent mechanical properties. The starting temperature of martensitic transformation (Ms) of the experimental steel was 389 °C and its critical cooling rate was 14 °C/s [19]. Therefore, under these experimental conditions, both the bainite transition and precipitation of the second-phase particles occurred. The bainite matrix provides high strength and mainly plays a strengthening role, while the high-density dislocations formed during the bainite transformation further contribute to the enhanced plasticity. The residual austenite increases plasticity during deformation, while the supersaturated C element in α-Fe can form Cr carbide second-phase particles during the aging treatment [27], which play a role in strengthening and plasticizing the steel.

We adopted the new C–Mn–Cr composition design scheme for the experimental steel, considering the effects of the alloying elements on the hardenability, solution strengthening, and the adjustment of the austenite transformation products. It replaced the Mn element in medium manganese steels, and the B element in hot-formed steels, with the Cr element to achieve microstructure control and improve hardenability. High strength and plasticity were achieved through microstructure design, diffusive reinforcement, and through the second phase particles replacing residual austenite.

### 3.2. Microstructure and Properties of Hot-Formed Parts

The mechanical properties of hot-formed parts from cold-rolled annealed plates are shown in Figure 5 and Table 3. It can be seen that the hot-formed specimens had good plasticity (elongation ≥ 11%) while maintaining high strength (≥1500 MPa). In the hot forming process, the experimental steel must be heated to austenitizing temperature. The microstructure obtained after cold rolling and continuous annealing changed completely due to the austenitizing process. Therefore, the enhanced plasticization mechanism of hot forming is different from that of cold-rolled annealing sheets.

The microstructure of the hot-formed parts is shown in Figure 6 and Figure 7. The figures show that the microstructure consisted of martensite (M), nanosecond-phase particles, and a small amount of residual austenite. The strength of the hot-formed parts was mainly achieved by martensite, while the plasticity was ensured by the second-phase particles and residual austenite. Although the strength of 22MnB5 is also due to martensite, its elongation is only 6% [28], which is much less than the elongation of this experimental steel. The experimental steel was Cr alloyed, on the basis of excluding the B element.

22MnB5 also has a fully martensite structure after hot forming. The experimental steel had a typical martensite structure in all three positions after hot stamping and forming. However, the experimental steel also had high elongation at higher strengths. In order to explore the mechanism of the high product of strength and elongation, the residual austenite in the hot forming of different positions was tested by XRD; the results are shown in Figure 8 and Table 4. It can be seen that the residual austenite fractions in the three positions of the hot-formed part were 4.26, 2.18, and 1.61%, respectively. The fraction of residual austenite in steel is related to the cooling rate: the higher the cooling rate, the smaller the fraction of residual austenite. The cooling rate of the bottom of the U-shaped specimen was the fastest due to the full contact with the abrasives and indenter, the cooling of the top part was the slowest, and the cooling of the side part was in the middle. There were thus great differences in the residual austenite fractions of the three parts. Although the residual austenite fractions of the three parts were different significantly, they were all less than those of 22MnB5. Such low residual austenite content is not sufficient to produce high plasticity.

It is well-known that grain refinement is an effective approach to simultaneously enhance strength and plasticity. In order to control the grain growth reasonably, a slightly higher temperature than the austenitizing temperature of 20Mn2Cr steel (the Ac_3_ of experimental steel is 879 °C [26])was selected as the holding temperature. The grain size of martensite ferrite, based on EBSD measurements, is illustrated in Figure 9, while the dependence of the average grain size in different positions is summarized in Table 4. The average grain size varied from 2.95 to 3.91 μm for angle boundaries larger than 10° [19], which is an approximate size level beneficial for both strength and plasticity [27].

The EDS analysis of second-phase particles distributed in the martensitic matrix is shown in Figure 10. It can be seen that they were mainly Fe carbide particles, with no Cr carbides. The analysis suggests that the original microstructure of the experimental steel before hot forming was granular bainite and ferrite, which is completely different from the ferrite and pearlite original microstructure of 22MnB5. The carburite in the microstructure was granular carbide and, under hot forming and holding (holding time was 3 min), this granular carburite was not completely dissolved. The cooling process creates the core of a new carburite-shaped nucleus. Insoluble carburite consumes the carbon content of austenite, which reduces the residual stress caused by shear during the transformation of austenite to martensite. As a result, the martensite strength was reduced but plasticity was increased. A large number of spherical carburites act as a second-phase particle reinforcement. At the same time, these second phase particles may alsoact as a potential plasticizer [27]. In summary, this interaction effect endowed the experimental steel with both good strength and plasticity.

Cr-alloyed steel has a beneficial effect on the surface oxidation resistance. It is good for the development of new, uncoated hot-stamping and forming technologies. The hot-formed surface topographies of 20Mn2Cr steel and 22MnB5 steel are shown in Figure 11. It can be seen that the thickness of the decarburization layer for 22MnB5 was 40 μm and the thickness of the decarburization layer for 20Mn2Cr was almost 0 μm. The oxidation resistance and decarburization properties of 20Mn2Cr were significantly better than those of 22MnB5 after heating to 910 °C, maintaining the temperature for 10 min, then hot forming. 22MnB5 is a typical hot-stamping-formed steel, used both in China and internationally.

Although 20Mn2Cr is an outstanding steel, characterized by a high product of strength and elongation and with great potential for use in the cold and hot stamping of automotive parts, there are also some challenges and tasks to be addressedin the future. For example, the elongation of the hot-stamped steel and the strength of the cold-rolling sheet are still not large enough. In future work, the elongation of the final hot-stamped product and the strength of the initial cold-rolling sheet should be improved through an investigation of the mechanism of strength and plasticity enhancement.

## 4. Conclusions


(1)A high product of strength and elongation was obtained for the automotive steels designed, allowing for both cold forming and hot forming using a multiphase, multidimensional reinforced plasticization concept. The products of strength and elongation for cold-formed and hot-formed steel were 20 and 18 GPa·% respectively, while the tensile strengths were more than 1000 and 1500 MPa, respectively.(2)The good mechanical properties, with regard to cold and hot forming, of this new automotive steel were due to an optimized alloy design. With the proposed methodology, a multiphase, grain-refined, and multidimensional microstructure of bainite, martensite, and nanosecond phase particles could be obtained under simple production process conditions.(3)This new automotive steel is characterized by its low-alloy design, simple production process, and good oxidation resistance.


## Figures and Tables

**Figure 1 materials-14-01121-f001:**
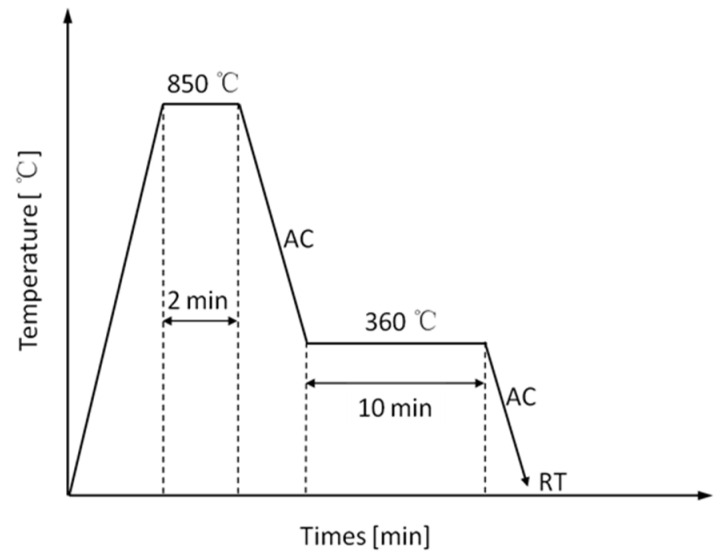
Schematic diagram of continuous annealing.

**Figure 2 materials-14-01121-f002:**
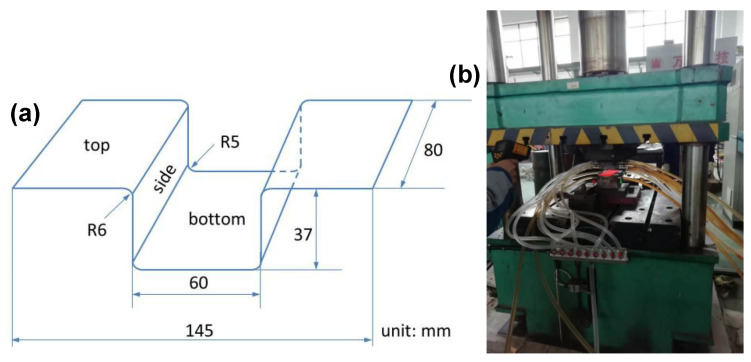
Schematic diagram of hot stamping of U-shaped part (**a**) and the shaper for U-shaped part (**b**).

**Figure 3 materials-14-01121-f003:**
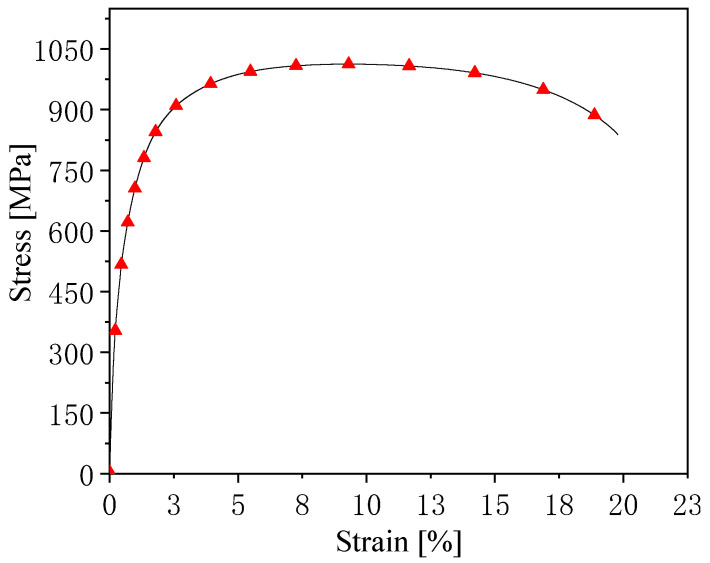
Engineering stress–strain curve of cold-rolled sheets after continuous annealing.

**Figure 4 materials-14-01121-f004:**
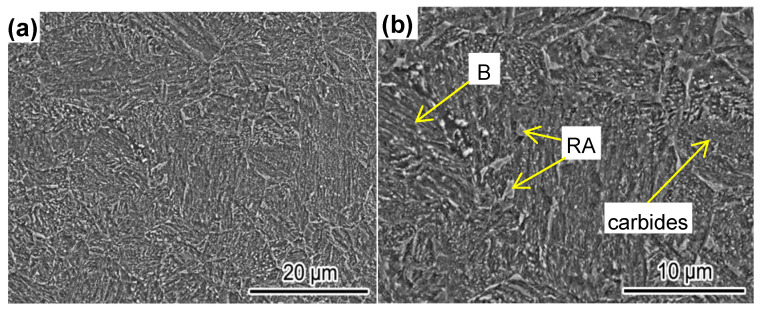
The microstructure of cold-rolled plates after continuous annealing. (**a**) Low amplification; (**b**) high magnification.

**Figure 5 materials-14-01121-f005:**
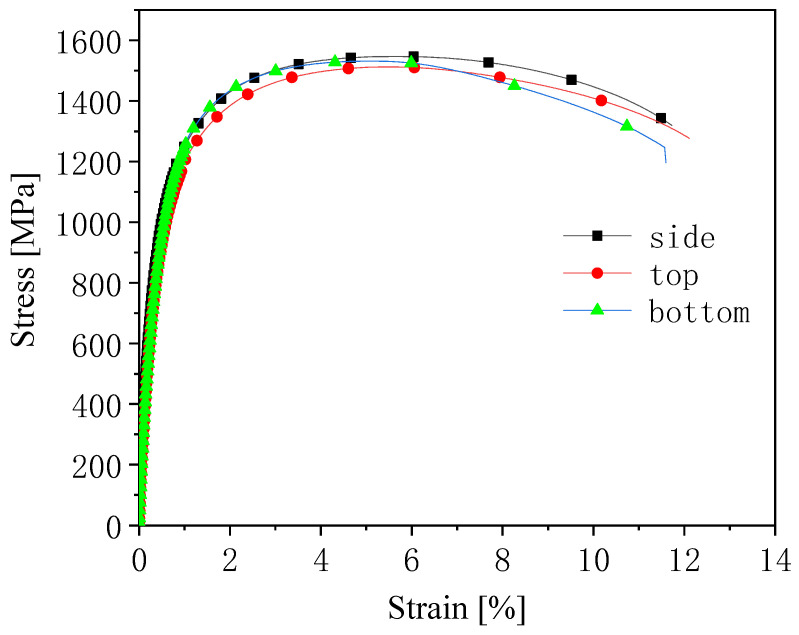
Engineering stress–strain curve of hot-stamping parts in different positions.

**Figure 6 materials-14-01121-f006:**
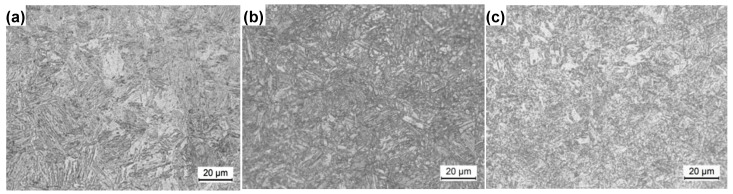
Microstructure of hot-formed parts in different positions: (**a**) top; (**b**) side; and (**c**) bottom.

**Figure 7 materials-14-01121-f007:**
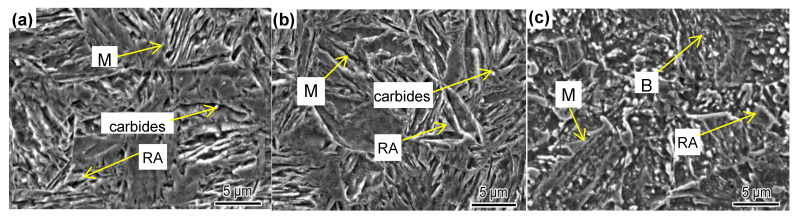
SEM of hot-formed samples in different positions: (**a**) top; (**b**) side; and (**c**) bottom.

**Figure 8 materials-14-01121-f008:**
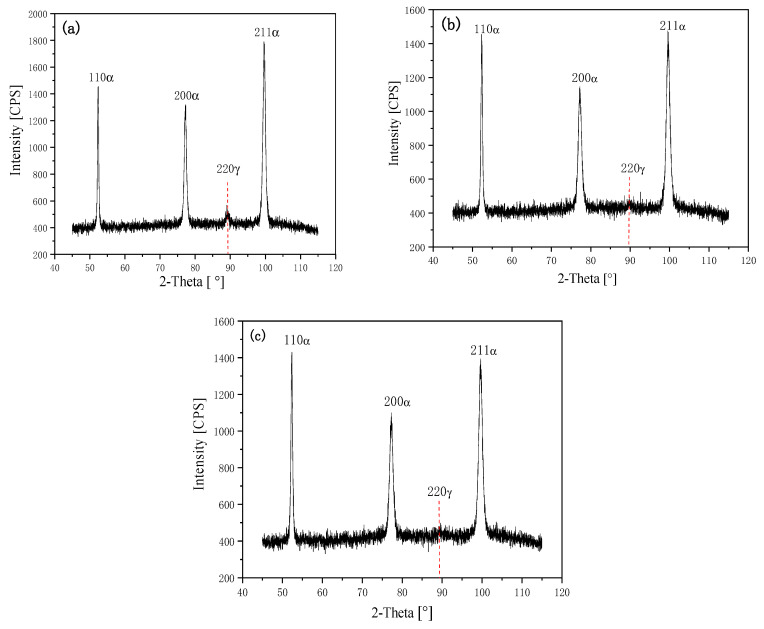
X-ray diffraction (XRD) spectra of hot-formed parts in different positions: (**a**) top; (**b**) side; and (**c**) bottom.

**Figure 9 materials-14-01121-f009:**
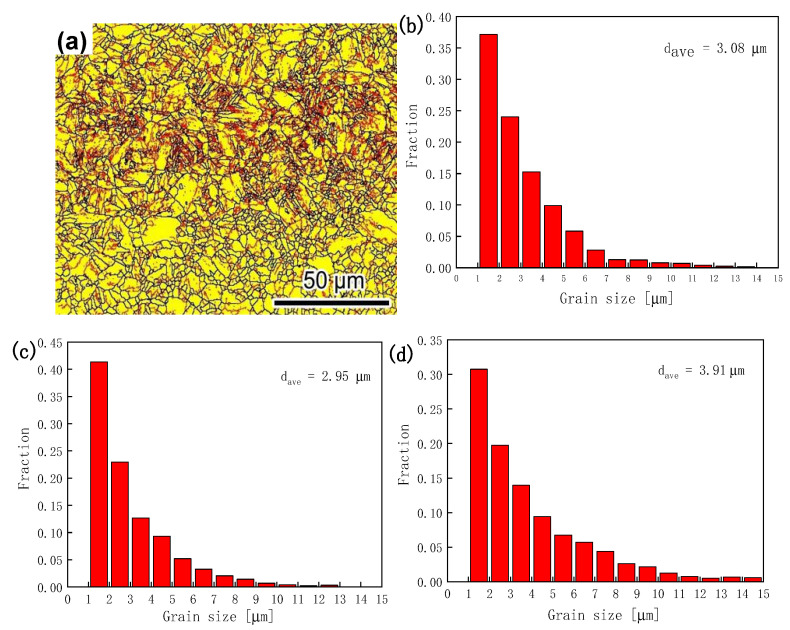
(**a**) EBSD images of grain size for hot-formed parts from the side position with high angle boundaries >10°; grain size distribution of the martensite ferrite for hot-formed parts in different positions: (**b**) top; (**c**) side; and(**d**) bottom.

**Figure 10 materials-14-01121-f010:**
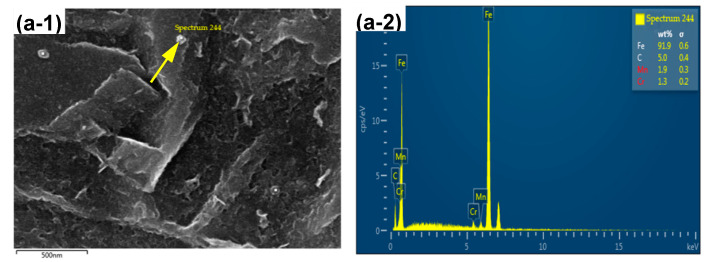
SEM and EDS of second-phase particles for hot-formed parts in different positions: (**a-1**,**2**) top; (**b-1**,**2**) side; and (**c-1**,**2**) bottom.

**Figure 11 materials-14-01121-f011:**
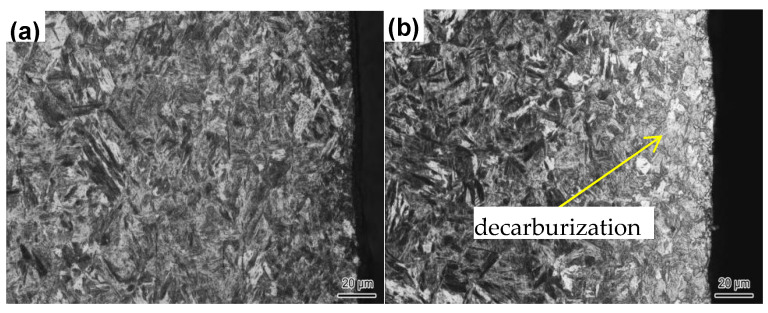
Surface topography of hot-stamped parts of 20Mn2Cr (**a**) and 22MnB5 (**b**).

**Table 1 materials-14-01121-t001:** Chemical composition of the new cold-rolling steels (wt%) [26].

Specimen	C	Mn	Cr	Nb	Si	Ti	S	P	Fe
20Mn2Cr	0.21	1.69	1.30	0.009	0.04	0.002	0.005	0.007	Balance

**Table 2 materials-14-01121-t002:** Tensile properties of cold-rolled sheets after continuous annealing.

Specimen	*R*_m_ [MPa]	*R*_p0.2_ [MPa]	*A* [%]	*R*_m_ × *A* [GPa·%]
20Mn2Cr	1013	726	20.0	20.3

**Table 3 materials-14-01121-t003:** Tensile properties of hot-stamping parts.

Specimen	Position	*R*_m_ [MPa]	*R*_p0.2_ [MPa]	*A* [%]	*R*_m_ × *A* [GPa·%]
20Mn2Cr	Top	1513	1190	12.0	18.2
	Side	1547	1285	11.5	17.8
	Bottom	1532	1280	11.5	17.6

**Table 4 materials-14-01121-t004:** Volume fractions of RA and grain size for hot-formed parts in different positions.

Position	RA (XRD)/%	Grain Size (EBSD)/μm
Top	4.26	3.08
Side	2.18	2.95
Bottom	1.61	3.91

## Data Availability

The data presented in this study are available on request from the corresponding author. The data are not publicly available due to the data also forms part of an ongoing study.

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
