# Peer review of "Automotive Steel with a High Product of Strength and Elongation used for Cold and Hot Forming Simultaneously"

_materials, 2021, doi:10.3390/ma14051121_

Round 1

Reviewer 1 Report

The paper deals with a problem of new grade steel which have high strength and also high elongation. The steel can be used for cold and hot forming processes to produce automotive components.

In the paper I found some inaccuracies that should be explained and corrected:

  1. The references used are almost sufficient for the paper's issues clarification. However each one (two) of the quoted references should be discussed individually and demonstrate their significance to the work. It is not necessary used four or even five references in one bracket: [6-9], [14-18].
  2. In the introduction we can see only one comparison new type steel to the 22MnB5 steel which characterized high strength and low elongation. Authors should add more information (strength and elongation) about TRIP and TWIP steel [1-18].
  3. There are no information about: dimensions of ingot, final temperature of forging process, type of rolling mills (hot and cold), schedule of hot and cold rolling process, hot rolling process: number of passes, initial and final temperature, etc.
  4. If the plate is 2 mm thick, it is sheet, not plate.
  5. Figure 2: add main dimensions
  6. What was the yield stress value of initial sheet and final product?
  7. Tables 3 and 4: authors should add some explanation, if the strength values for different parts of the U-shaped profiles are almost the same (Table 3) and also grains size are almost the same why there are high differences in the residual austenite fractions in the three parts?
  8. What was the initial value of the residual austenite fractions?
  9. Fig. 10. In the description of the Figure, there is no a), b) and c)
  10. Fig. 11. What was a thickness of the decarburisation layer?
  11. There are many mistakes, for example: line 48: …performance. the hot forming…, line 53: …steel plate, It will often…, line 55: Inorder…, line 65: …newcomposition…, line 80: …production), The process…, line 82: (see Figure2.), line 89: …at 20kV, The corresponding…, line 97: …ofcontinuous…, and so on.

Author Response

 please see the attachment!  thank you !

Reviewer 2 Report

In this paper, a new chemical composition of automotive steel is proposed. The developed automotive steel  have excellent properties, such as strength and elongation. The new elaborated steel can be used for both cold forming and hot-forming to produce high-performance automotive components. The article is very important for automotive industry. However, the current form of the article requires some corrections:

  1. Please check, if the presented results are correct, e.g.
  • differences between Rm - Figure 3 and Table 2 for the elongation 19,7 % [line 113])
  • differences between Rm - Figure 5 and Table 3 [line 148, 150])

If the results are correct, please comment on the differences

  1. How to interpret the coefficient (Rm×A) adopted by you. When the strength increases and elongation decreases, we can obtain a similar value as in the opposite case. Moreover, the results of Rm×A  - Table 3 are rounded up once  or sometimes down. Why?
  2. Line 107: These results are similar to those of commercial QP980 – please confirm (please refer to the literature or your own results in this field)
  3. Line 77: Table 1. Chemical composition of the new cold rolling steels (wt%) - Please, provide source of presented data
  4. The microstructure was observed with ZEISS observer A1m metallurgical microscope and JSM-6610 scanning electron microscope – please explain, what the observer A1m means
  5. Figure 7 and Figure 9. – please correct the description of scale
  6. Please read the article carefully and correct any spaces and typos (I am giving only examples below):

Line 197: s 10min (incorrect) in line 116: 10 min (it is correct)

Line 117: he microstructure of

Line 118: production)are

Line 121: °C/s[25]

Line 128: anneal- ing[26]

Line 72: 130kg

Line 54: 1000MPa

Line 38: conditions[12

Line 29: materials[6

Line 89: 20kV, The corresponding

Line    104 :  are shown in Figure 3 and Table 2. they can be see that the tensil

Line 107: are 1013MPa, 19.7% and 20.1GPa·%

Figure3.Stress

Line 142: on≥11%

Line 160: s (a)top, (b)side, (c)bottom

Line 176: 879℃[28])of

Author Response

please see the attachment!  thank you !

Reviewer 3 Report

The manuscript was about a high-strength steel with large plasticity and included interesting results as well as well explained discussion. My comments are below;

What is the future challenges of 20Mn2Cr steel? Or are there no more tasks to be solved? There was no mention about the negative points of 20Mn2Cr in the manuscript. If 20Mn2Cr is a perfect steel, then every steel should be replaced by it right now. Please add some comments at the end of the discussion.

"holding pressure was 20t" (line85) ==> What does the "t" unit mean? Ton? Torr? Please use SI unit.

Both "hotformed parts" and "hot formed parts" were used. Please unify the expression.

The phrase "In order to explored the mechanism of high product of strength and elongation." (line166) has two mistakes. First, "to explored" was grammatically wrong. It should be "to be explored" or "to explore". Second, there was no verb, so this phrase is not a sentence. 

"At the same time ***. At the same time, ***." (line203-204) Please do not use "at the same time" twice.

In addition, there were many mistakes (some are listed below). English editing is requred.
"theresidua" (line12)
"quench-parts(QP)" (line44)
"currently .For" (line52)
"plate, It" (line53)
"Inorder to" (line55)
"newcomposition" (line65)
"plate, After" (line74)
"production), The" (line80)
"ofcontinuous" (line97)
"Table 2. they" (line104)
"can be see" (line104)
"production)are" (line118)
"maintaining  high" [Double spacing] (line142)
"and1.61%" (line169)
"879oC[28])of" (line179)
"shown in the figure. 10." (line193)
"a potential At" (line203)
"Figure11" (line214)

Author Response

please see the attachment! thank you !

Reviewer 4 Report

Dear authors.

The idea and concept of steel presented in this scientific work will be interested in the automotive industry.  But on the other hand, the manuscript consists of a lot of grammar mistakes and inaccurateness. This work must be improved.

Comments:

- Pleas description in the text of the manuscript the physical basis of the calculated parameters of steel with units GPa*%. Line 15, 106.... What this parameter is mean?

- The text of the manuscript consists a lot of grammar and stylistic mistakes and inaccurateness (small letter in the beginning sentence, spaces between words, join words ….). Please carefully read your manuscript and correct it. Line: 48, 51-54, 65, 74, 97, 117, 163,  193, 235…..and other.

- Please explain why the experimental steel was designed with the presented chemical components? 

- All graphs in the manuscript must be the same definitions of units and on the axis. For example, Fig. 1 is Time/ min but Fig.5 Strain, % components?

- Please describe how was prepared samples for EBSD?

- The results presented in Fig. 3 and 5 must be specified: is it true or engineering stress–strain curves?

- Table 2. 1,013GPa*19,7% it is not 20,1GPa*%.

- The scale in Fig. 4a and 4b mast be improved. These scales on both pictures are not understandable.

- The scale in Fig. 7 and 11 must be also improved.

- The capital letters M, B, RT on the microstructure must be described in the text of the manuscript.

Author Response

please see the attachment ! thank you !

Round 2

Reviewer 1 Report

The paper is ready for publication

Reviewer 4 Report

Dear authors.
I haven’t got any comments for the new version of the manuscript. This work could be accepted for publication.

Best regards.